# The Need for Better Attachment Bonds Between Institutional Caregivers and Children in Residential Care: A Systematic Review

**DOI:** 10.3390/bs15030245

**Published:** 2025-02-20

**Authors:** María-Jesús Martínez-Usarralde, Raquel Conchell, Mónica Villar, Lucía Pérez-Tabernero

**Affiliations:** Department of Comparative Education and History of Education, Faculty of Philosophy and Educational Sciences, University of València, 46010 Valencia, Spainlucia.perez-tabernero@uv.es (L.P.-T.)

**Keywords:** attachment, institutional caregiver, institutionalized children, review, PRISMA method

## Abstract

Attachment is an emotional bond based on the first relationships formed between people. In the case of institutionalized children, it is more difficult to establish a secure attachment bond with an adult. The figure of the institutional caregiver is key to promoting secure attachment and emotional support for these vulnerable children. This paper examines children in residential care (0–17 years), their relationship with attachment, and the figure of the institutional caregiver in relation to attachment. We conduct a systematic review of the scientific literature (SLR) carried out using the PRISMA method. The results suggest that institutionalized children are more likely to have behavioral problems because of the type of attachment they develop, and that institutional caregivers mediate attachment through their competence. In conclusion, institutional caregivers are an essential support figure in the life of residential care centers; because of this, these professionals must have adequate working conditions and receive ongoing training and support. Also, it would be necessary to analyze the appropriate skills of institutional caregivers to make appropriate intervention with the minors.

## 1. Introduction

Attachment is highly relevant to the personal and social construction of children institutionalized in residential care, making the figure of the institutional caregiver a powerful mediator. Research on the impact of attachment on children’s development determines how they internalize relationships with others and, therefore, how they will socialize in the future ([1]; [4]; [16]; [11]; [25]). Traditionally, little attention has been paid to attachment theory among children in residential care, so there is a growing demand to focus not only on improving living conditions in institutions, but also on ensuring children’s physical and emotional well-being. This line calls for a more holistic approach that recognizes the importance of individualized care, connection with the community, and the integral development of children, also from the more social, emotional, and affective side ([2]; [5]; [7]; [19]; [10]; [11]; [18]; [20]; [21]). It is essential for institutional caregivers to be accessible referents with whom children can form a bond, so that we can provide institutionalized children with a safe and caring environment that promotes the development of healthy attachment relationships. Their contribution to emotional and affective care can have a lasting impact on the emotional and social well-being of these vulnerable children ([8]; [12]; [15]; [31]; [27]).

Attachment theory ([1]; [4]) refers to the relationship established from birth (or even before) and during the first years of life with a person’s primary caregiver. It determines how children understand relationships with others and, therefore, how they will socialize in the future ([11]; [25]). The theory emerged from the need to provide answers and explanations for how people develop psychobiological, representational, and relational strategies to regulate stressful experiences ([16]).

In the early years, children create their mental representations of relationships based on the example of their primary caregiver. The loss or separation from their attachment figure, according to [25] ([25]), often has repercussions in adult life, ranging from problems in their ability to regulate and manage their emotions and behaviors, to the adoption of more aggressive and impulsive behaviors, as well as conduct and behavioral disorders ([9]; [11]; [13]). Research on the developmental consequences of child abuse and neglect highlights the central role of attachment in reversing developmental risks and provides opportunities for prevention and early intervention. It highlights the need to address these issues from a lifelong developmental perspective, from infancy through adolescence to adulthood, in order to gain a better understanding of how certain phenomena affect different stages of human development. Considering these changes over time contributes to a more holistic understanding ([32]).

### 1.1. A Snapshot of Children in Residential Care

According to [11] ([11]), institutionalized children have not had the opportunity to grow up with their parents or caregivers and are dependent on an institution or child protection services. The child protection system in Spain and other countries consists of a set of social policies and legal regulations that propose measures, procedures, principles of action, resources, and professional practices to guarantee children’s rights, with society as a whole—and specifically the public authorities, i.e., the administration—being directly responsible for their protection ([33]).

Regarding residential care, [26] ([26]) argues that despite being the least appropriate resource for situations of child homelessness, it is still the most widely used, compared to foster care and adoption. Above all, we need to understand that the transfer of a minor to a residential institution is an important transition, which may result in feelings of abandonment, leading to risk and vulnerability for them ([8]).

Regarding the social imagination of institutionalized children, several systematic reviews and meta-analyses have shown differences between children in institutions and those living with their biological families. The former are more likely to have insecure ([6]; [17]) or particularly disorganized attachment patterns ([30]; [34]; [35]), as well as poorer social skills related to values of cooperation, assertiveness, self-control, and responsiveness ([24]). According to [18] ([18]), attachment is essential for reducing difficulties and negative influences in the development of children’s strengths (e.g., prosocial behavior).

### 1.2. The Professional Value of the Institutional Caregiver

Research shows that in residential care, physical presence is not enough for a child’s social and emotional development ([5]). It is, therefore, essential that all professionals in caregiver/educator roles are selected and trained in attachment theory, motivation for the role, teamwork skills, childcare, empathy, conflict management skills, emotional disposition, listening skills, and knowledge of child development and children’s rights ([3]).

[8] ([8]) point out that children’s relationships with caregivers in residential care are fundamental. Establishing a good future attachment,] and, therefore, optimal relationships, depends on previous experiences and relationships—as we have already mentioned, most importantly with their caregivers—and the type of bond they have created. Thus, we can say that secure attachments play a crucial role in the holistic development of children in residential care. According to the available evidence, the establishment of a secure attachment can create an essential bond that contributes to reducing the incidence of developmental difficulties in children in this situation ([18]).

The first pillars of social functioning lie in the quality of attachment bonds formed with caregivers. As with other aspects of social functioning, disturbances in the formation of attachment relationships are particularly common among children raised in institutions, as children in these circumstances have few opportunities to form lasting relationships with secure and caring adults ([10]). It is important to highlight and emphasize the important role of social educators or institutional caregivers in residential care, even promoting the figure of the “attachment tutor” ([15]).

## 2. Methodology

### 2.1. Desing

The SLR followed the Preferred Reporting Items for Systematic Review and Meta-Analysis (PRISMA) guidelines, and the study protocol was prospectively registered in the International Prospective Register of Systematic Reviews ([23]).

### 2.2. Search Strategy and Screening

Regarding the search strategies, we carried out a systematic search in the following electronic databases: Dialnet (https://dialnet.unirioja.es), Scopus (https://scopus.com) and Web of Science (WoS, https://webofscience.com). With respect to the language of the publications, we performed the search in Spanish and English, and only documents published between 2015 and the beginning of 2023 were analyzed. We supplemented the results with retrospective searches to identify relevant publications. Articles were managed using Endnote version X9. We carried out the search strategy in an iterative process, using multiple combinations of the research project keywords, including the following terms: “institutionalized children”, “juvenile center”, “social education”, “institutional caregiver” and “attachment”, with the Boolean operators “OR” and “AND”.

In terms of eligibility, the inclusion criteria were the following: (1) research published in peer-reviewed journals to ensure that included studies met minimum methodological standards; (2) papers analyzing the relationship between attachment and children in residential care; (3) papers identifying the relationship between social educators or institutional caregivers and attachment; (4) papers on the role of social educators or social caregivers in the attachment of institutionalized minors; (5) research published in open access journals; (6) research in the fields of social sciences, psychology and education; and (7) papers published between 2015 and early 2023.

To identify studies that met the eligibility criteria, two reviewers independently scanned the abstract and title of each article. Two authors independently assessed the full text of selected items, and discrepancies were discussed and agreed with the other authors. We contacted the study authors by e-mail when we needed additional information for our review or to determine eligibility. Two of the researchers extracted the data independently. The selected studies met the agreed inclusion criteria. Disagreements were resolved by consensus with a third author.

We used the Mixed Methods Appraisal Tool (MMAT; [14]; [22]) to assess the methodological quality of the studies. For each study design, the MMAT provides a checklist of five questions to evaluate methodological quality. Response options include “yes”, “no” or “can’t tell” (if the study did not provide the necessary information for a clear answer). Three authors independently assessed the risk of bias in the trials. Disagreements were resolved by discussion and consensus with the participation of another author.

The database search yielded 192 articles (see Figure 1). After removing duplicates, 123 items remained. Further papers were excluded based on titles and abstracts, and 33 papers were selected for full-text analysis. Of these, 18 articles were excluded because they did not meet the stated criteria. In the end, a total of 15 papers met the requirements.

In the risk of bias results, we assessed the risk of bias in the included studies using the MMAT criteria (*n* = 15; see Table 1). In terms of data, the sample presented in most studies (*n* = 12) is highly or fairly representative of the target population, except for three studies that did not meet this condition (studies 3, 4 and 11). When considering whether the studies provide adequate assessments (or exposition) of the outcome and intervention, we see that ten of the studies adequately meet this criterion, and only five need some improvement (studies 2, 4, 6, 8, 11). Regarding the completeness of outcome data, only one (study 11) did not provide this information. If we focus on whether the articles consider confounding factors in their design and analysis, almost all of them meet this criterion (*n* = 12), except for some limitations that show room for improvement (in studies 2, 4 and 11). All studies completed their study periods as planned.

### 2.3. Conceptual Definition Generation Procedure

The findings are based on a qualitative analysis of the abstracts and results of each of the 15 selected documents (see Table 2). In terms of the results of the studies included (all of which analyzed the relationship between attachment and children in residential care), we found that most of the research emphasizes the importance of attachment in the lives of these children, based on the importance of attachment bonds, which the studies relate to different variables. Insecure attachment is known to be more common in institutionalized children and can lead to various problems in the future, such as conduct disorder and psychosocial risk behavior. Conversely, a secure attachment with an adult facilitates emotional and social skills and can lead to better self-esteem ([2]; [5]; [7]; [19]; [10]; [11]; [18]; [20]; [21]).

## 3. Results

The simple indicators relate to three research variables: gender, year of publication, and authorship. In terms of gender, there are more women publishing research on this topic. The percentage of men is much lower, at 16.7%, while 83.3% of the authors are women. As regards the year of publication, we can observe that most of the documents analyzed were published in 2019 (33.3%), followed by 2015 (20%). It should be noted that there is not much literature combining the three selected variables (“attachment bonds”, “institutionalized minors”, and “social education”) before 2015. Looking at the authors who signed the papers, we can see that most of them are not recurrent names, except for Matos, Mota and Costa, in that order, who are more productive authors in this field of research, signing between two and four papers each.

With regard to the content indicators, the following variables were selected: type of center, sample, age of the sample, and whether the research was quantitative or qualitative. Firstly, regarding the type of center where the research was carried out, the majority are residential care centers, although there are also two centers for the execution of judicial measures. Secondly, about the subjects included in the sample, the majority (66.7%) were children, followed by the group of educators or carers (33.3%). Thirdly, about the age of the children, most of the studies focus on 6-, 7- and 12-year-olds, while the least studied groups are 1- and 11-year-olds. Finally, regarding the type of methodology used in the research, half of the studies are purely qualitative (including two literature reviews), while 42.9% are quantitative (using different scales) and 7.1% use a mixed design.

Regarding the results of the selected studies that analyzed the relationship between attachment and the figure of the social educator or institutional caregiver, we found no direct relationship between the functions of social educators and attachment. However, all the studies emphasized the need for social educators or institutional caregivers to be emotionally accessible for a secure attachment to be established. They also consider a good attachment bond with an institutional caregiver as a facilitator of emotional skills, interpersonal skills, and self-image ([3]; [15]).

Considering the results of all these studies analyzing the relationship between social educators or institutional caregivers and children institutionalized in care centers, it is imperative that these vulnerable children have a good relationship with their educators or institutional caregivers. To this end, their role (as social educators or institutional caregivers) should be strengthened so that they can be considered professional parental figures ([29]). We should also bear in mind that the factors influencing the bond formed between them are individual and emotional, rather than contextual, which means that it is essential that reference people provide multidisciplinary and high-quality care for children. Most empirical studies highlight the importance of the relationships developed between caregivers and institutionalized children in mediating positive adjustment to the institution as well as improved well-being and resilience. Attachment plays a fundamental role in this. Therefore, it is important to professionalize the role of social educators and provide better training to these institutional caregivers so that institutionalized children can develop in the most satisfactory way ([8]; [12]; [15]; [31]; [27]).

## 4. Discussion

This paper has looked at children institutionalized in residential care and their relationship with attachment. We found a few studies that examined these two interrelated variables, mainly because of the difficulty in accessing this type of sample. Numerous studies from around the world have shown the importance of attachment in the present and future development of children. It is essential that international and national institutions work to implement attachment rehabilitation programs to prevent difficulties. For this to happen, it will be necessary to continue research along the same lines to better understand the relationship between attachment and difficulties for children in residential care.

There is no doubt that the lack of research on this subject is one of the current challenges in the field, as there is almost no research on the relationship between attachment and the figure of the social educator or institutional caregiver. Future lines of research should, therefore, analyze the relationship between these two variables, since it is clear that the construction of the bond is an essential element in attachment.

Social educators or institutional caregivers need to develop networks of care and emotional support for these vulnerable children. However, current research suggests that there are professionals who do not know how to deal with this and even assume that there is no need to discuss emotional support, contact, listening, and care. This is partly because there are not enough professionals who can accompany children in their development and adequately address the emotional needs of each child. Therefore, decent working conditions are essential. On the other hand, skills and training are fundamental, as is the understanding that these institutions can and do involve family dynamics and routines. Thus, the group under study expects permanence, regularity, and consistency in their care, all aspects that can facilitate affective reorganization. Caregivers need to develop attitudes that allow them to connect with these children and validate their emotional experiences, providing support that goes beyond behavior management and promotes the development of parenting skills. Having resources, skills and strategies for self-regulation is ultimately fundamental to establishing an effective emotional connection with children.

The limitations encountered during the study had to do mainly with the small amount of bibliography found in open access, which greatly limited the subject. On the other hand, an important limitation has been that the figure of the institutional caregiver in childcare centers is different in each country.

Finally, if we look to the future, it would be interesting to carry out research that compares what we have analyzed in this literature review, and therefore, what the authors say, with the reality of the children’s centers (most of them privately administered), to see if institutional caregivers are becoming secure attachment figures in children’s centers and how to improve this attachment.

## Figures and Tables

**Figure 1 behavsci-15-00245-f001:**
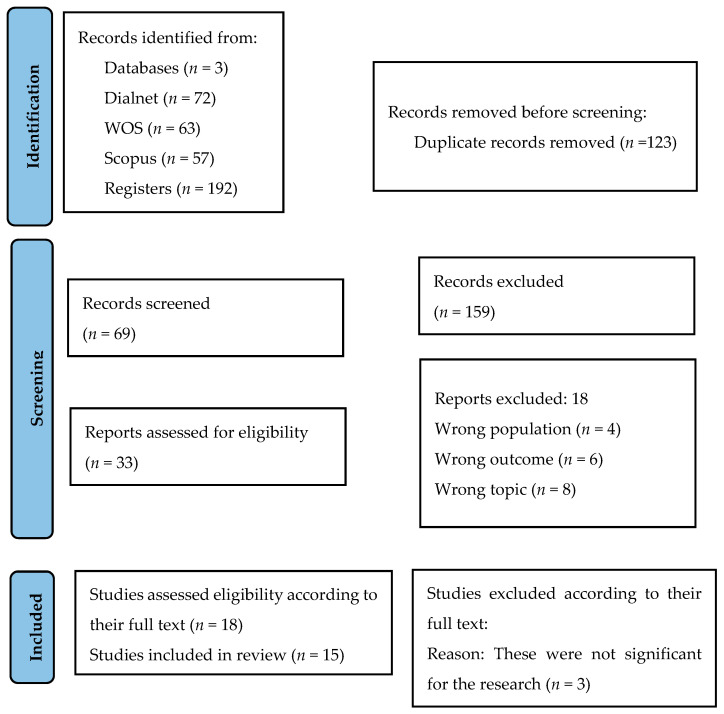
Flow diagram.

**Table 1 behavsci-15-00245-t001:** Risk of bias of included studies.

Study Number	Authors	Are the Participant Representative of the Target Population?	Are Measurements Appropriate Regarding Both the Outcome and Intervention (or Exposure)?	Are the Complete Outcome Date?	Are the Confounders Accounted for Un the Design and Analysis?	During the Study Period, Is the Intervention Administered (or Exposure Occurred) as Intended?
(1)	[3] ([3])					
(2)	[8] ([8])					
(3)	[18] ([18])					
(4)	[2] ([2])					
(5)	[10] ([10])					
(6)	[21] ([21])					
(7)	[12] ([12])					
(8)	[20] ([20])					
(9)	[19] ([19])					
(10)	[31] ([31])					
(11)	[15] ([15])					
(12)	[11] ([11])					
(13)	[5] ([5])					
(14)	[28] ([28])					
(15)	[7] ([7])					


 Inadequate 
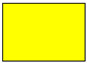
 Some improvement 
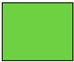
 Appropriate

**Table 2 behavsci-15-00245-t002:** Selected papers.

Title	Journal	Authors and Year	Variables	Database
lnstitutional caregiver experiences in child care.	Ese Anna Nery	[3] ([3])	Social education; lnstitutionalised children	Web Of Science
Predictors of the quality of the relationship with caregivers in residential care.	Children and Youth Services Review	[8] ([8])	Social education; lnstitutionalised children	Web Of Science
¿Qué papel tiene el apego en la aparición de dificultades y fortalezas en menores en acogimiento residencial?	Psychology, Society, & Education	[18] ([18])	Attachment; lnstitutionalised children	Web Of Science
Early Maltreatment and Current Quality of Relational Care Predict Social- Emotional Problems Among lnstitutionalized lnfants and Toddlers.	lnfant Mental Health Journal	[2] ([2])	Social education; lnstitutionalised children	Web Of Science
Signs of attachment disorders and social functioning among early adolescents with a history of institutional care.	Child Abuse Negl.	[10] ([10])	Attachment; lnstitutionalised children	Web Of Science
Caregivers’ Attachment and Mental Health: Effects on Perceived Bond in lnstitutional Care.	Professional Psychology: Research and Practice	[21] ([21])	Attachment; Social education; lnstitutionalised children	Scopus
"Getting involved": A thematic analysis of caregivers’ perspectives in Chilean residential children’s homes.	Journal of Social and Personal Relationships	[12] ([12])	Social education; Institutionalised children	Scopus
Adolescents in lnstitutional Care: Significant Adults, Resilience and WellBeing.	Child Youth Care Forum	[20] ([20])	Attachment; Social education; lnstitutionalised children	Scopus
The Role of Attachment in the Life Aspirations of PortugueseAdolescents in Residential Care	Child and Adolescent Social WorkJournal	[19] ([19])	Attachment; Social education; lnstitutionalised children	Scopus

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
