# Peer review of "The Need for Better Attachment Bonds Between Institutional Caregivers and Children in Residential Care: A Systematic Review"

_behavsci, 2025, doi:10.3390/bs15030245_

Round 1

Reviewer 1 Report

Comments and Suggestions for Authors

Dear Authors,

the suggestions for some changes/improvements can be found directly in the comments of the document of your paper "The need for better attachment bonds between institutional caregivers and children in residential care: A Systematic Re-view".

Author Response

Comment 1: Please briefly explain what each color in the table means. Although the text preceding the table explains it, when looking at the table it is still unclear what the red, yellow and green colors mean
Respose 1: We agree with the reviewer's comments. We have added a legend - page number 5, paragraph 2 and line 207-209.

Comment 2: Please could you clarify what the limitations of this review are and what recommendations should be made for future studies on this topic. What should be emphasized as particularly important for improving attachment support for children in institutional care?
Respose 2: As indicated in the conclusions we have clarified what the limitations of the review and future recommendations are. And it highlights issues to be considered for improving attachment support for children in institutionalized care. - page number 7, paragraph 2-5 and line 276-308.

Comments 3: Add a concluding paragraph with summary ideas
Respose 3: We have added in conclusions a last paragraph that synthesizes the conclusions of the work. - page number 7, paragraph 6 and line 309-313.

Comment 4: Please check again the referencing style given in the instructions especially for italic text.
Respose 4: We have corrected all references with the proper style. - page number 8, paragraph 6 and line 324-408.

Reviewer 2 Report

Comments and Suggestions for Authors

It is a very interesting work, with a reliable and precise methodology although it does not lead to results and discussion. Specifically, I hope to be able to contribute with the following recommendations:

Abstract

It would be recommended to present the sample with basic data, such as the sex of the minors and age (mean, interval, for example).

Based on the "Instructions for Authors" of this journal, it would be desirable to explicitly state the conclusions of the study.

 Introduction

It would be advisable to explain the importance of attachment in general before narrowing down the study population. It is narrowed down specifically, generally, and then specifically again and might seem disorganized.

There is a lack of data on the child's previous attachment, whether the state of vulnerability of having to separate from his or her family poses a risk to attachment, beyond global implications. In the event that there is no literature, it would be interesting to reflect this, since damage to primary attachment poses a barrier to bonding.

It would be appropriate to expose the difference in attachment by demographic variables such as age or sex, since they moderate much of the related literature.

Following the "Instructions for Authors" of this journal, it would be necessary to include objectives and hypotheses. Specifically, "It should define the purpose of the work and its significance, including specific hypotheses being tested".

Methods

Regarding the design, greater specification would be desirable.

It would be important to specify the characteristics of the samples with respect to age and sex for each study separately, in line with indicating the results in this regard in the abstract. For example, 66.7% were male children.

Results

Since the hypotheses or objectives of the study are not clear (in the introduction), there is also no clarity in the results regarding what is sought to be defined in the review. It is not stated whether the review is classic (therefore aimed at answering a specific question) or more of an exploitative nature that seeks to present a vision on the subject. I recommend readjusting and exploring this issue in greater depth, since the lack of coherence or purpose of this review detracts from the Methodology.

Discussion

The discussion is not developed per se, since it is not contrasted with previous studies. Even though there are few studies, it can and should be compared with them, like attachment in other populations. Specifically, based on the "Instructions for Authors" of this journal: "Authors should discuss the results and how they can be interpreted in perspective of previous studies and of the working hypotheses. The findings and their implications should be discussed in the broadest possible context and limitations of the work highlighted. Future research directions may also be mentioned. This section may be combined with Results".

Conclusion

This section is not mandatory but can be added to the manuscript.

References

There is no reference from recent years (2023 y 2024).

Author Response

Comments 1: In the abstract it would be advisable to present the sample with basic data, such as the sex of the children and the age (mean, range, for example).
Respose 1: We have added as indicated in the abstract the age of the children - page number 1, paragraph 1 and line 15.

Comments 2: It would be useful to explain the importance of attachment in general before delimiting the study population. It boils down to the specific, the general and the specific again, and may seem disorganized.
 There is a lack of data on the child's previous attachment, whether the vulnerable state of having to be separated from his or her family poses a risk to attachment, beyond the global implications. In the absence of literature, it would be interesting to reflect this, as damage to primary attachment poses a barrier to bonding.
Respose 2: As the reviewer indicates, a small explanation of attachment in general has been added. There are no data on attachment risk in institutionalized children - page number 2, paragraph 2 and line 36-39.

Comments 3: Following the “Instructions to Authors” of this journal, it would be necessary to include objectives and hypotheses. Specifically, “the purpose of the work and its significance should be defined, including the specific hypotheses being tested”.
Respose 3: We understand the reviewer's indications, since it is a systematic review, it is indicated that hypotheses are not necessary.

Comments 4: Following the “Instructions for authors” of this journal, it would be necessary to include objectives and hypotheses. Specifically, “the purpose of the work and its significance should be defined, including the specific hypotheses being tested”.
Respose 4: We understand the reviewer's indications, being a systematic review it is indicated that hypotheses are not necessary.

Comments 5: The discussion is not developed per se, since it is not contrasted with previous studies. Although there are few studies, it can and should be compared with them, such as attachment in other populations. Specifically, based on the “Instructions for Authors” of this journal: "Authors should discuss the results and how they can be interpreted in perspective of previous studies and working hypotheses. The findings and their implications should be discussed in the broadest possible context and limitations of the work should be highlighted. Future research directions may also be mentioned. This section can be combined with Results.
Respose 5: As stated in the conclusions, we have clarified what the limitations of the review and future recommendations are. And highlighted issues to consider in improving attachment support for children in institutionalized care. - page number 7, paragraph 2-5 and line 276-308. And we have added in a final paragraph that synthesizes the conclusions of the paper. - page number 7, paragraph 6 and line 309-313.

Comments 6. There is no reference to recent years (2023 and 2024).
Respose 6: As the reviewer indicates there is no reference to 2023 and 2024 since the study ended in the middle of 2023.
